# OpenReview forum: "Cooperation, Competition, and Maliciousness: LLM-Stakeholders Interactive Negotiation"
_NeurIPS.cc/2024/Datasets_and_Benchmarks_Track — NeurIPS 2024 Track Datasets and Benchmarks Poster_

### Official Review · Reviewer_Dn58 · 2024-07-12
**A benchmark for quantitatively evaluating the negotiation skills of (AI) agents, the connection to LLMs and the focus on the benchmark can be improved..**

**Rating:** 4
**Confidence:** 3

**Review:**

The proposed benchmark focusing on negotiation is motivated by various real-world use cases where multiple agents need to interact with each other in a strategic way (namely cooperatively and/or competitively, or even adversarially). In this regard, a well-defined and implemented benchmark is useful in its own right. However, the connection to AI, or specifically LLMs does not seem as well motivated.

For instance, in abstract:
> We create a testbed of complex multi-agent, multi-issue, and semantically rich negotiation games.

It does seems that, such testbed is significant by itself, and does not need LLMs.

In lines 29-30:
> A future where AI assistants communicate on behalf of different entities seems plausible. This raises the concern of models being exploited by rogue parties to pursue unaltruistic or manipulative goals.

Even without AI assistants, the concern of being exploited by rogue parties persists. So this motivation for the need to consider some form of robustness specifically in AI agents-based negotiation is not so clear.


Subsequently, this emphasis on LLMs may take away the strength of the contribution because of the shifted focus, which should be on the benchmark, in this dataset and benchmark track. For instance, one conclusion from the empirical results is that GPT-4 is better at GPT-3.5 in various settings, but this conclusion does not seem to directly analyze or evaluate the benchmark itself. Furthermore, in the proposed formalization (Section 2 for the set-up and Section 3 for the interaction protocol), it does seem that some simpler agents (e.g., rule-based, heuristic-based) seem reasonable built-in baselines that one might expect with such a baseline. Such simpler baselines can provide a meaningful gauge of the empirical results scored by the LLM-based agents. Currently, while there are quantitative results, it is not so clear how ``good'' these results are: How much room is there for improvement? Do LLM-based agents perform significantly better than the simplest possible heuristic-based agents to justify their complexity and higher costs?


In a brief summary, the proposed benchmark does have real-world motivational use cases, but the exposition and development can be made more extensive by structuring the (empirical) investigation centered around the benchmark itself and possibly including more exploration of alternative/baseline agents so that a practitioner using the benchmark (e.g., someone who designes a new algorithm/agent) can know how well the new algorithm performs against the baseline.

**Strengths:**

- There are real-world motivational use cases for the proposed benchmark.
- The formalization of the negotiation game and interaction protocol of the agents is clear.
- The empirical investigation is relatively extensive.

**Additional Feedback:**

In lines 50-51
> Furthermore, GPT-4 agents can get higher rewards compared to GPT-3.5 [5] ones when assigned the same role in a mixed population simulation, hinting at potential fairness and disparity considerations when users use models with varying capabilities as assistants.

The costs of the GPT-4 and GPT-3.5 should also be factored in, and a quantitative consideration should be used. Otherwise, the fairness aspect seems incomplete: A more expensive agent (GPT-4) being more capable than a less expensive one (GPT-3.5) does sound fair, though the quantitative cost-benefit ratio can provide more practical insight.

**Clarity:**

The paper is reasonably clear, but there are some clarification questions below:

In lines 37-40
>  Such skills are required in many applications to rank and propose solutions, e.g., to answer “ﬁnd the cheapest, shortest ﬂight with a reputable airline that will not lose my luggage". We ﬁrst use a role-play exercise commonly used for teaching negotiation [38], which consists of
 multiple parties and issues (see Figure 1).

What is the relevance of the example of finding a suitable flight to the negotiation scenario?

In line 44
> We use an LLM as a seed to design ...

What is meant by a "seed"?

In lines 48-49
> Our ﬁndings show that GPT-4 [29] (the best-evaluated model) still underperforms when increasing games’ difﬁculty.

What is GPT-4 compared against?

In line 64
> Games consist of 6 parties, ... , and 5 issues

Why specifically 6 and 5?

In line 86
> Each party $p_i$ has a minimum threshold $\tau_{p_i}$ for acceptance

Is the threshold meant for a deal?

In line 119
> where BATNA is Best Alternative To a Negotiated Agreement, which is usually the threshold $\tau_{p_i}$.

Why is there a utility if no agreement is reached at all? If some agreement is reached, then the best should correspond to a utility, right? so what is the motivation for BATNA?

In line 130
> Therefore, $S^*$ is $p_i$’s estimate.

How is this estimate constructed?

In line 168
> Therefore, as long as the agent has a next turn, we instruct it to generate a secret plan of possible next actions.

What is the motivation for making this part (i.e., secret plan) explicit? as in explicitly stored in some variable?

**Correctness:**

For this benchmark, the evaluation method can be made more focused (e.g., elaborate clearly why Sections 4.3 & 4.4 are meaningful evaluation of the benchmark) and more extensive (e.g., possible to expand Sections 4.6 and 4.7). I understand that investigating how different LLMs perform with/against each other in different settings/variants is an interesting setting, but it seems that the obtained results provide a evaluation for the LLMs (agents) rather than the benchmark itself.

**Documentation:**

There is sufficient detail (in the submitted source code) to support reproducibility.

**Ethics:**

No.

**Limitations:**

It seems that the authors note that the limitations are the set of considered methods (which are mostly LLM-based), setups and defenses. These seem like considerations of an algorithm design (such as for a negotiation agent, or a policy for defending against potential adversaries) instead of limitations of a benchmark. Perhaps the authors could clarify why these are the limitations of a benchmark, or point to where they may have provided such elaboration.

**Opportunities For Improvement:**

- The focus on the benchmark itself can be improved, such as by structuring the empirical investigation to be more centered on the benchmark and its different aspects, such as usability, maintainability, ease of extension and etc; and expand the set of metrics (such as those in Section 4.1) and elaboration on these metrics in terms of how a score of each metric can be interpreted, because these metrics are a key part of the benchmark itself; and further investigation and elaboration of the tunability of the difficulty of the game itself (such as its degree, interpretation).
- It may be helpful to include some simple agents as baseline, since the formalization in Section 2 & 3 does not seem to exclude such agents.

**Relation To Prior Work:**

The relation to prior work seems adequately discussed to me.

**Summary And Contributions:**

The authors propose a benchmark for evaluating the negotiation skills of AI agents, testing their arithmetic, inference, exploration and planning capabilities. They pay particular attention to large language models (LLMs) as the "agents" for the implementation and empirical investigation. The authors describe a formalization of the negotiation game, consisting of components including agents, "issues", scoring functions and a solution space, which is to be "played" in a multi-round way and can have different characterizations such as compromising game, greedy game, and adversarial game. They authors also empirically investigate the performance of LLM-based agents in their proposed benchmark to draw insights, such as the more advanced GPT-4 beats GPT-3.5 and the effects of a greedy and/or adversarial agent. Their results suggest that the proposed benchmark may prove difficult even for the most advanced LLMs (i.e., GPT-4) from their somewhat unsatisfactory performance.

---

> ### Author Rebuttal · Authors · 2024-08-14
>
> Thank you for your review and questions. We reply to them below.
>
> **- Motivating the benchmark:**
>
> - Thank you for acknowledging the significance of our benchmark.
> - Our motivation in using LLMs are two folds 1) LLMs are already used in many real-world tasks that require some form of negotiation (processing legal contracts [1]) and customer service chatbots [3], 2) There is a lack of dynamic multi-turn benchmarks that are easily extendable and adjustable in difficulty to test future models and combat memorization and leakage of benchmarks in training data.
> - Evaluating attacks (deliberate exploitation) or cases where models steer off their developers’ negotiation goals (e.g., prices, offers, discounts) becomes important now as there are already examples of such incidents [2,3]. Our benchmark and metrics for example evaluate cases where models propose deals that are less than a corresponding threshold.
>
> [1] https://www.luminance.com/news/press/20230511_luminance_announces.html
>
> [2] https://fortune.com/2024/07/16/negotiating-chatbot-nibble-ai-ecommerce/
>
> [3] https://www.bbc.com/travel/article/20240222-air-canada-chatbot-misinformation-what-travellers-should-know
>
> ----
>
> **- Baselines:**
>
> - We now add some heuristics and random baselines. Please see the global rebuttal. The most capable model in our experiment performed considerably better than the baseline in most cases.
> - Nevertheless, we note that given our motivation outlined above, the benchmark is valuable by itself to evaluate general capabilities of LLMs (e.g., arithmetic calculation, theory-of-mind, etc.) that are also useful for other tasks.
>
> -----
>
> **- Evaluating the benchmark itself:**
>
> - Thank you for your suggestions on highlighting different properties of the benchmark. We are happy to improve this part. **We believe most of the benchmark’s properties (e.g., extensions) are addressed within our experiments already and we will restructure and expand when needed**.
> - Nevertheless, **we note that our empirical experiments are designed to show the benchmark’s properties**. For example, highlighting that GPT-3.5 performs significantly worse is important to show that the benchmark is challenging.
> - Additionally, discussing performance under prompt variations shows evidence for how the benchmark can help evaluate these negotiation properties (e.g., if agents correctly infer the properties of others, etc.).
> - We believe giving extensive baselines in our experiments and discussing future planning algorithms, attacks, and defenses are strong contributions to holistically show the numerous aspects of how this benchmark can be utilized in the future as a simulation environment. As we don’t collect a new dataset (rather we design a simulation environment), evaluating LLMs on the benchmark, in our opinion, becomes integral and meaningful to evaluate the benchmark itself.
> - We also believe that our benchmark and experiments **are highly in scope within this track** (we kindly refer to the CfP, to quote: "benchmarking tools", "Systematic analyses of existing systems on novel datasets yielding important new insight", "Data generators and reinforcement learning environments.").
> - Following your suggestions, we are also happy to add more visualizations and statistics on, e.g., number of deals for each agent, etc.
>
> ----
>
> **- Limitations of the benchmark itself:**
>
> - We will expand our discussion on limitations. As per our reply to reviewer XyTr, we will discuss directions on how to set scores and thresholds to give more granularity and control.
>
> ----
>
> **- Comments on clarity:**
>
> We answer questions below and will clarify in the paper:
>
> - *“Flight examples”:* this is to show that the skills needed for negotiation are also needed for many other applications. To answer this, the agent needs to compare flight numbers (arithmetic calculation), understand users’ priorities (theory-of-mind), and weigh different options across different categories (price, schedule, airline reputation) to reach a final answer.
>
> - *“Seed”:* this refers to that we manually rewrite the game (and potentially by further queries to GPT4) to ensure logical consistency (as described in this paragraph).
>
> - *GPT-4 comparison*: we refer to comparisons across games that have comparable levels of difficulty (i.e., number of feasible deals) or across different variants (e.g., cooperative vs greedy).
>
> - *"Number of agents/issues":* please refer to our reply to reviewer iQ4e. As mentioned in our paper, we build on negotiation literature for teaching negotiation and systematically adapt it to evaluate LLMs by designing the metrics, interaction, and how to extend the benchmark.
>
> - *“Threshold”:* yes, this refers to thresholds on the scores of deals
>
> - *BATNA*, in negotiation literature, means the party's alternative if negotiations are unsuccessful.
>
> - *“Estimate”:* our model is generic to allow different ways of how it can be constructed. For example, in our baseline framework, this can be based on observing other agents’ interactions or common sense and background information of the game.
>
> - *“Secret plan”:* we are not sure we get the question. The motivation is to carry over the plan as a form of self-generated feedback or a step in a chain-of-thought way.
>
> ----
>
> **- Additional feedback:**
>
> - *“Fairness”:* thank you for raising an interesting point. Cost can be a direct way of comparison between GPT-4 and GPT-3.5. However, we believe the comparison is less clear between open-source models. Even if we consider the computation required given by the scale of models, it is less clear as well since some smaller or comparable models may have better performance, and therefore users may not be clearly informed about the capabilities.
>
> -----
>
> We hope our response has addressed your concerns and will be glad to answer and incorporate further comments.

---

### Official Review · Reviewer_iQ4e · 2024-07-25
**Review comments of Submission 1007**

**Rating:** 7
**Confidence:** 3
**Correctness:** Yes.
**Clarity:** Yes.

**Review:**

This benchmark is somewhat useful for validating the reasoning and arithmetic capabilities of LLM agents. However, it falls short in assessing their negotiation capabilities in realistic scenarios, thereby lacking practical value.

**Strengths:**

S1: This benchmark is highly extensible, allowing for easy control of the game's difficulty and scenarios through a set of parameters.

S2: This benchmark effectively tests the capabilities of LLM agents in inferring others' preferences and combinatorial optimization, which are crucial for negotiation.

S3: The experiments are extensive, and various ablation studies demonstrate the effectiveness of each prompt within the benchmark.

**Additional Feedback:**

N/A

**Documentation:**

Yes.

**Ethics:**

No.

**Limitations:**

Yes.

**Opportunities For Improvement:**

W1: This benchmark appears to oversimplify negotiation scenarios and lack scalability. The games are limited to only six players and five issues, with each issue containing only three to five options. The authors have not justified these ad hoc settings and have not explained whether such configurations can be easily extended to larger-scale negotiations.

W2: A fundamental factor in real-world negotiations is the irrational behavior of players. However, the authors set the temperature of the LLM agents to 0 and did not assign them different personas, turning these negotiations into purely rational games. Consequently, this benchmark may fail to evaluate the ability of LLM agents in real-world negotiations.

W3: Modeling negotiation problems as a combinatorial optimization problem aimed at maximizing weighted social welfare is not convincing. Intuitively, whether it is a compromising game, greedy game, or adversarial game, the players' negotiation objective should be to maximize their own expected payoffs. Any compromise with others' interests is merely a means to increase the likelihood of reaching an agreement, thus serving as a variable that influences their own expected payoffs. Therefore, the authors should provide theoretical or empirical justifications to show it is a real problem.

**Relation To Prior Work:**

Yes.

**Summary And Contributions:**

This paper introduces a benchmark composed of "scoreable negotiation games" to test the negotiation capabilities of LLM agents. In these games, LLM agents engage in multiple rounds of negotiations across several issues, with the final agreement requiring a selection from various options under each issue. Each agent has different private scores for each option, and the agreement must meet everyone's minimum score threshold. To complete this benchmark, LLM agents should be able to infer the private scores of others through interaction and solve a combinatorial optimization problem based on these inferred scores to maximize some weighted social welfare. Although this benchmark appears to assess negotiation capabilities at a high level, some settings are oversimplified and lack sufficient justification, making it unconvincing as a principled benchmark.

---

> ### Author Rebuttal · Authors · 2024-08-14
>
> Thank you for your review and acknowledging the strengths of our work. We reply to your concerns below:
>
> **- Scalability:**
>
> Please find our comment on constrained action space in the global rebuttal. In addition, we would like to clarify:
>
> - **Our work is the first to extend LLM negotiation benchmarks to more than two players**. The total number of available deals is considerably large (**720** possible combinations). We also want to note that we tested variants of the game by generating custom versions using GPT-4 in Section 4.5 and different levels of difficulty, making our benchmark highly **extensible**.
>
> - As mentioned in our paper, our design choices are based on established negotiation literature which we adapt to evaluate LLMs in a structured and comprehensive way. We note that this simulation exercise (as noted by Susskind et al.) usually takes 3 hours for human participants to complete, indicating that it is not trivial. It was previously reported that very few participants reached unanimous agreement [1]. Such game simulations have also been widely adapted in negotiation skills training since then due to their nuanced nature. However, they are usually only qualitatively analysed with focus on the lessons learned by the participants rather than assessing the achieved scores to judge their performance as we have systematically done for the LLM agents in our benchmark.
>
> -----
>
> **- Irrational behavior:**
>
> - We kindly note that the experiments with temp. “0” is not a property of the benchmark itself. The benchmark can be very easily run with other models and other hyperparameters.
>
> - We also believe that the roles and background of agents are ways of modulating personas (e.g., one agent is exclusively pro-environment, while others care about the environment, but still consider other factors).
>
> - In addition, we also tested greedy and adversarial settings which can be considered as different personas (c.f. Section 3.2). In fact, the adversarial agents can arguably be seen as irrational agents (at least wrt other agents’ perspectives) since they may propose deals that are not aligned with common sense and may lead to low scores for themselves as a potential sabotaging technique. Such variants can also be used to study robustness against noisy irrational strategies.
>
> - We believe our benchmark would be valuable to extend to other personas inspired by behavioral game theory, and study other aspects that possibly affect negotiation as biases.
>
> -----
>
> **- Modelling negotiation:**
>
> - We believe that our purely cooperative game can also be helpful to study collaborative problem solving under unknown variables (the scoring of other agents) and it works as a baseline to evaluate agreement in the easiest setup to establish models’ performance. Even in this setup, agents have to non-trivially iteratively refine their strategies based on the feedback received from other parties.
>
> - Our formulation and experiments allow flexibility in varying the level of cooperation. We also now add an experiment where all agents primarily maximize their payoffs (please refer to the global rebuttal).
>
> - Importantly, we would like to note that we assume a non-zero sum game where there may exist deals A and B such that A Pareto-dominates B. I.e., the player has an incentive in switching strategy when a different strategy has potential to reach agreement (e.g., by giving higher scores to veto parties) even if its scoring function does not improve (as already seen several times in the qualitative analysis). While this might seem like a social welfare maximization, it’s actually a reflection of interdependent utilities in the proposed negotiation settings.
>
> - Finally, we kindly note that our model for negotiation games is inspired from literature (as in Susskind et al.) and is utilized in negotiation skills training, suggesting its efficacy and realism in simulating realistic problems.
>
> [1] https://confengine.com/conferences/agile-games-2016/proposal/2118/test-your-multi-team-negotiation-skills-with-harvard-harborco

---

> > ### Author Rebuttal · Authors · 2024-08-20
> >
> > Dear reviewer iQ4e,
> >
> > We would like to further illustrate our point of the non-zero sum game and how our formulation does not contradict with real world negotiation with two examples. The first is based on the scores of the game and the second from observed qualitative analysis of agents' interaction.
> >
> > In the attached pdf in Fig 1, we iterate over all possible deals' combinations and compute their scores with respect to $p_1$. For each possible deal, we also compute the number of agreeing parties. **For each value of $p_1$'s score, there may exist many possible deals that would map to that value. These will have different levels of agreement based on others' scores**. Therefore, we plot on the x axis $p_1$'s scores and on the y-axis the number of agreeing parties across all deals that map to that particular value of $p_1$'s score. As can be observed, at the same achieved utility, there can be deals that satisfy more agents. Therefore, in order to increase the success likelihood, $p_1$ would have incentive to shift strategies, especially if it would achieve the same utility. This is consistent with our formulation of optimizing both utility and agreement. Also, please note there may be deals with both higher utility and higher agreement (please see the max agreement values between scores of 70-80). This analysis is similar to Figure 11 in the appendix.
> >
> > In addition, we show in Fig 2 one example from agents' interactions (an agent other than p1). Here, the previously proposed deal (still at early rounds of the negotiation) would give the agent a perfect score of 100 (maximum utility). However, since the agent is prompted to collaborate (optimize for agreement *as well*), the agent first makes observations and inferences about other preferences, and eventually adjust the deal slightly. Even though the agent is instructed to be cooperative, the finally suggested deal is still significantly above that agent's minimum threshold (50), empirically suggesting (along with many other qualitative examples) that cooperative LLM agents, in our setup, optimize for both agreement and utility. As more observations are added with more interactions, the agent could further compromise, also ideally optimizing for both objectives.
> >
> > We hope these examples further clarify our setup and motivation. We will add this discussion and examples to the paper.

---

> > > ### Comment · Reviewer_iQ4e · 2024-08-23
> > >
> > > Thank you for your detailed explanation and justification. I have two more questions. First, why was the number of players set to six, and can we increase it to accommodate large-scale negotiations? Second, if the temperature is increased to 1, would there be any significant changes in your findings/conclusions?

---

> > ### Author Response · Authors · 2024-08-24
> >
> > Thank you for the questions. We have run two experiments:
> >
> > 1- We used 6 players following negotiation literature of established negotiation exercises [1]. We now add another game where we prompt GPT-4 to add one additional player and one additional issue while specifying the motivation and preferences of that additional player and the preferences of the original players wrt the new issue. The total number of deals is now **2880 vs 720** used originally. We manually set the scores to have a comparable ratio of feasible deals (~7%) and we ran GPT-4 with this new game. We ran this experiment 80 times to accommodate the larger action space.
> >
> > | Any success | Final (6-way) | Final (7-way) | Wrong deals |
> > | ---------------   | ----------------- | ------------------ | ---------------- |
> > | 96%              | 63%               |     18%          |      0%           |
> >
> > Thank you for the suggestion which we believe would further enhance the diversity of our benchmark and show its scalability to potentially more players and issues in the future to simulate even larger scale negotiation. We will include this game in the paper and our code.
> >
> >
> > 2- We added an experiment for the base game and GPT-4 with temperate 1 (computed over 35 simulations)
> >
> > | Temperature | Any success | Final (5-way) | Final (6-way) | Wrong deals |
> > | ---------------   | ----------------- | ------------------ | ---------------- | ----------------- |
> > |         0           | 100%            | 81%                |     33%         |      1%           |
> > |         1	       | 96%              | 68%                |     6%           |      0%           |
> >
> > Compared to temperate 0, the performance drops in some metrics, especially the final deal, although it’s significantly higher than the random chance (please check the baselines in the global rebuttal). The (any success) metric is more stable since it’s computed for any deal made by p1 during the session so this may counter the randomness. We believe this would not change the findings as the general trend still holds (e.g., LLM agents, using powerful models, are better than random chance and show good performance in metrics such as arithmetic calculations of deals, etc). Nevertheless, as you noted, this can be another interesting follow-up variant of the evaluation to study robustness under randomness when all agents (or a subset) are irrational.
> >
> > [1] Susskind et al. "Using simulations to teach negotiation: Pedagogical theory and practice." Teaching negotiation: Ideas and innovations (2000): 285-310.

---

> > > ### Comment · Reviewer_iQ4e · 2024-08-24
> > >
> > > Thank you. I appreciate the additional experiments and have changed my rating accordingly.

---

### Official Review · Reviewer_kvGq · 2024-08-07
**Good benchmark for LLM-based multi-agent systems**

**Rating:** 7
**Confidence:** 3
**Correctness:** Yes
**Clarity:** Yes

**Review:**

I enjoyed reading the paper. It is clearly exposed and well written. Even though benchmarks for LLMs in multi-agent environment already exist (see e.g., https://arxiv.org/pdf/2402.16499), I believe that the settings describe in this paper are a good addition to the current existing ones. Furthermore, the ability to progressively increase the difficulty of the benchmarks, as well as the ability to easily generate semantically different games, makes these benchmarks promising to be relevant for newer and better versions of LLMs.

**Strengths:**

- Novel benchmark to evaluate LLMs interacting with each other.
- Good ablation study of the LLM's prompt to determine the effect of each part of the prompt's structure.
- Exhaustive comparison across closed and open source LLMs.
- Adaptive difficulty of the benchmark.

**Additional Feedback:**

Is it possible to analyse the outcomes of the negotiation games with standard game theory concepts such as Nash equilibrium or Pareto optimality?

**Documentation:**

- There is not much documentation on the GitHub repo apart from basic commands in readme files on how to run the benchmarks.
- The way the code is organized makes it difficult to use and extend. There are multiple notebooks with similar names and it is not clear what they are showing. Overall needs to be better organized and packaged.

**Ethics:**

All good.

**Limitations:**

- The proposed games only allow agents to interact via text thus it is not possible to compare the performance of LLM-based agents with other kinds of agents based on other methods (RL, for example).

**Opportunities For Improvement:**

- I gave a quick look at the GitHub repository. the readme could be a bit more elaborate with small examples on how to run the benchmarks quickly.

**Relation To Prior Work:**

- I think a mention of this stablished paper would be good: https://arxiv.org/abs/2304.03442 given that it concerns interacting agents with LLMs
- Potentially also relevant: https://arxiv.org/pdf/2312.03664

**Summary And Contributions:**

This paper introduces a novel benchmark for evaluating LLMs in the context of multi-agent negotiation scenarios. The benchmark consists of negotiation games where LLMs are prompted with the game instructions and history as context and guided to observe, explore, and plan their actions. The authors introduce numerical metrics to evaluate the performance of LLMs as agents and show the performance of several SOTA models. The benchmark can be successfully made harder to pass by requiring higher scores for deals to be successful, thus making it a useful benchmark to progressively asses the improvement of LLMs in these tasks.

---

> ### Author Rebuttal · Authors · 2024-08-14
>
> Thank you so much for your encouraging review!
>
> - We are happy to expand our related work section as we outlined in the global rebuttal to include broader context beyond LLM negotiation.
>
>
> - Thank you for the suggestions on documentation. We plan to improve our documentation and code as we mentioned in the paper’s submission checklist to make it easier to extend to new scores and games. The public Github repository dates back to when the paper first appeared online with an initial version and we will update it with the new experiments and additions.
>
>
> - **Limitations:** While we propose text-based games to evaluate LLMs, the games can potentially be used for other methods by communicating via deal suggestions and other actions that are not purely text.
>
>
> - **Questions:** Please refer to our reply to reviewer XyTr where we discuss how to potentially use Pareto optimality principles to design the scores and thresholds of games. We leave further detailed analysis and investigating adaptation of game theory and behavioral game theory to future work as these interesting questions require more investigation. We believe our work can possibly be positioned in an intersection between auction theory (e.g., designing the format and rules and the private value assumptions of scores) and cooperative game theory (e.g., studying coalitions) and exploring the parallels between these fields and our negotiation problem can provide great insights for future work.

---

> > ### Author Response · Authors · 2024-08-24
> > **We improved our code and documentation to enable easier extension**
> >
> > Dear reviewer kvGq,
> >
> > Thank you again for your suggestions. We would like to let you know that we have now updated our code following your suggestion (https://github.com/S-Abdelnabi/LLM-Deliberation).
> >
> > - We provide an easier setup to change games and support changing number of agents and issues easily. We have added "scores files" to easily instantiate new games by changing scores.
> > - We have now already expanded the game to more players and issues using this new setup.
> > - We added documentation on how to instantiate games with different combinations of: models, incentives, etc.
> > - We also added a guide on how to change incentives in the future.
> > - We added more documentation on how to run the evaluation.
> >
> > Please also note that we plan to create a page with scoreboard and main results as mentioned in our paper's checklist.

---

### Official Review · Reviewer_XyTr · 2024-08-08
**Cooperation, Competition, and Maliciousness: LLM-Stakeholders Interactive Negotiation**

**Rating:** 7
**Confidence:** 3

**Review:**

The paper presents a well-implemented benchmark for evaluating language models' capabilities in multi-agent negotiation scenarios with a limited action space. While it could be expanded, the quality of the research high, with clear definitions, formalized methods, and reproducible results. The benchmark itself is dynamic and capable of assessing current and future models. This work provides a valuable tool for evaluating and comparing the negotiation capabilities of different language models.

Quality and Clarity: The paper is well-written and logically structured. The authors have done an excellent job formalizing the deliberation process, clearly defining concepts throughout. The results are presented clearly, with tables and figures that effectively illustrate the performance of different models and game variants.

Originality: The benchmark's design, incorporating three game formats, is novel and provides insights into different negotiation dynamics. The use of chain-of-thought prompting and the inclusion of a scratchpad for strategic planning add additional depth and context to the experiments.

(more details below)
Pros:
- Well-implemented, reproducible benchmark
- Clear formalization of negotiation concepts and methods
- Novel multi-agent scenario with six agents
- Flexible difficulty settings for future adaptability
-  Incorporation of chain-of-thought prompting and strategic planning elements
- Interesting results, particularly in comparing different game variants

Cons:

- Lack of baseline comparisons (random and heuristic-based agents)
- Limited qualitative analysis of negotiation strategies
- Missing empirical results for all-greedy agent scenarios
- Constrained action space may limit generalizability to more open-ended scenarios

Overall, despite its limitations, this work meaningfully contributes to the field of multi-agent language model evaluation. The benchmark provides a solid foundation for future research in this area, although extrapolating the results to more open-ended negotiation scenarios should be limited. The addition of baseline comparisons, all-greedy scenarios, and more qualitative analysis would further strengthen the paper's contributions.

**Strengths:**

- The code and game implementation for this benchmark is well executed. It's easy to use, follow, reproduce, and increase in difficulty, making it applicable to increasingly capable models. The code caters to both open and closed source models, and the accompanying materials include a quick start guide for reproducibility.
- The incorporation of chain-of-thought (CoT) structure throughout the experiments, along with various prompt ablations (Table 1), strengthens the methodology significantly. The authors' approach to testing robustness against semantically similar changes by prompting GPT-4 to rewrite the base game while maintaining semantic relationships is commendable.
- The formalism behind the experiment design, including the use of BATNA, is clearly explained. The benchmark's use of 6 agents is an advancement over previous work with fewer agents, providing a more complex multi-agent scenario.
- The results in Table 5, showing that the presence of one greedy party yielded a lower success rate (57\% for 5/6-way agreement) than the presence of an adversarial party (63\% for 5/6-way agreement in the untargeted scenario), are particularly intriguing.
- The benchmark's flexible difficulty, adjustable through fewer players and lower minimum scores, allows for assessment of future, more capable models.
- The inclusion of mixed model variations (Table 2) and prompt ablations (Table 1) improves the quality and robustness of the evaluation.

**Additional Feedback:**

I am excited to see more benchmarking in the multi-agent space. With the incorporation of the changes recommended above, I would consider upping the overall score.

**Clarity:**

The paper is well-written and structured clearly. The figures and tables effectively support understanding of the approach and results. There are a couple of small language changes I would recommend (see above).

**Correctness:**

The scenarios are based on negociation literature and are logically constructed with appropriate testing. The claims and methodology are sound, and the evaluation metrics lead to appropriate conclusions based on how the benchmark was designed. Please see my comments above for additional improvements here. The high reproducibility is commendable.

**Documentation:**

The documentation and publicly available code are of extremely high quality, which greatly enhances the paper's contribution. It is ethical, sufficiently detailed, and user friendly.

**Ethics:**

The paper appropriately discusses the small potential ethical implications around adversarial scenarios. I do not have any concerns on this.

**Limitations:**

The authors could expand more on the limitations of the prescribed action space and scoring system. While they mention the benchmark can be made more difficult, more discussion of how the constrained format may or may not generalize to real-world negotiation would be valuable.

**Opportunities For Improvement:**

- A random baseline for comparison would be valuable to contextualize model performance relative to chance. This is even more important because action space is extremely constrained. I looked into the prompts in the appendix and there is a limited possible number of deal combinations, meaning it is important for readers to know how often randomly compatible deals could be made. If you tested this and it was indeed extremely low, I recommend mentioning that.
- The process for setting minimum thresholds for each agent is not well explained. These appear to be arbitrary scores out of 100 that differ for each agent. You acknowledge that these can change to impact the difficulty of the benchmark, but there may be better ways to more granularly measure that. For example, a pareto efficiency calculation of potential deals (720 possible deals based on napkin math on the prompts) and the distance between the selected deal and the optimal one. The minimum thresholds being manually set is not a dealbreaker, but it will likely be important to explore more granular measurements in the future. It may even allow you to integrate more rigid benchmark conditions (like profile-based rigid minimum thresholds) based on negotiation/debate literature.
- The rationale for requiring 5/6 agents to meet the minimum threshold for a successful deal in the compromising (and probably greedy) environment, rather than unanimous agreement, is not clearly explained. This choice meaningfully impacts the benchmark's difficulty. I suspect it is because this decision enables comparison with the Adversarial environment or because the drop-off in performance with all 6 parties may also be a factor, though please correct me here. Related to this - further discussion on performance changes with different numbers of agents (beyond the results in Table 4) or additional experiments playing with game lengths would be insightful as this impacts scalability.
- I felt that a scenario where all agents are performing in a 'Greedy' manner was missing. Given the emphasis on real-world tasks and the interesting results with one greedy agent (Table 5), this seems like a valuable experiment that could have been implemented.
- While the quantitative results are thorough, there's limited qualitative analysis of the negotiation strategies employed by the models. It shows up inconsistently, particularly in the GPT-3.5 analysis and Greedy/Adversarial experiments (Table 5 and Figure 5), but is not systematic. Example “Another party, p2, provides a budget for the project and has 68 veto power. It usually acts as a middle ground between different parties.“ Is this spontaneous or deliberately prompted?
- Minor point - in both the abstract in introduction, you say "GPT-4 and SoTA large models (e.g., Llama-3 70b) still underperform." It is unclear what this means. Underperform in comparison to what? There is no human, random, or heuristic baselines for comparison.

**Relation To Prior Work:**

The authors clearly outline how this work extends previous research and have cited relevant papers. However, more connections to work on debate, negotiation, game theory, and deliberation from fields could provide valuable context and further inform both metrics and additional environment design.

**Summary And Contributions:**

This paper presents a benchmark for how LMs can deliberate over a proposal and arrive at common ground. The authors have developed a set of mixed-motive, non-zero-sum negotiation games where multiple LM-backed agents try to reach an agreement (BATNA) on various issues. The authors built several negotiation games where LM backed agents enter a turns based schema where one party ($p_1$ for the final deal) proposes a solution to the subset of issues ($I$), and that is either rejected or accepted by each individual agent in accordance with their own minimum threshold ($\tau_{p_i}$). In order to reach a consensus, 5/6 models must have their minimum thresholds met by the proposal.

There are three game formats: Compromising - where all agents are looking to compromise and are not prioritizing their own reward score. Greedy - where one or more agents are trying to maximize their own scores while aiming for an agreement. Adversarial - where one agent is trying to sabotage the whole deal through targeting. The models were provided with a conversational history and a scratchpad to enhance observation and strategic planning. The settings were sufficiently hard so that GPT-3.5 and below level models were unable to consistently arrive at a deal. GPT-4 and SOTA open-source models were able to arrive at common ground with varying degrees of success.

---

> ### Author Rebuttal · Authors · 2024-08-14
>
> Thank you very much for your time and the very thorough, encouraging and constructive review! We appreciate your feedback and we are happy to incorporate it to improve the work! Please find comments and answers below.
>
> **- Baselines, All-greedy experiment, scaling to less-constrained setups, updating related work discussions:**
> - Thank you for the suggestions. Please refer to the global rebuttal for details.
>
> -----
>
> **- Qualitative analysis:**
> - We will add a more detailed appendix. We will provide a more systematic qualitative analysis by first identifying properties and behaviors that are common/required in negotiation (e.g., similar to Susskind et al.) and give examples to where that was observed (or lacking) in the outputs of agents. For example, we can check how 1) agents infer the priorities of others and understand their own preferences, 2) compromise for less important issues, 3) observe opportunities for building coalitions, 4) observe if consensus on issues have reached, 5) find valid options, 6) find supporting argument, and 7) finally communicate that in their public answers while ensuring not to communicate secret scores and plans.
>
> - We will give examples for these dimensions across the different game variants highlighting any differences. For example, in greedy games, agents may refuse deals that do not give them their highest option (e.g., the highest union quota for the local union agent) and propose others that contain the best option but would give a comparable or even slightly less overall score. We will also give examples highlighting when adversarial players succeed or fail in sabotaging agreement. Given the limited space of rebuttal, we are happy to include the details in the final version.
>
> -----
>
> **- Minimum thresholds:**
>
> - As mentioned in the paper, we took inspiration from negotiation literature (Susskind et al.). We adapt their game to evaluate LLMs by proposing the communication protocol and the metrics. The base game’s background and scores are adapted from theirs. We write our descriptions and we also change the setups (proposing the different game variants) and propose to change the difficulty by changing the thresholds. For other games, we aimed for comparable difficulty by having a similar number of solutions. However, this was a mainly manual process by tweaking and observing the set of feasible deals and the number of feasible deals for each agent. We acknowledge that this is a limitation of our work.
>
> - An alternative option is to more explicitly algorithmically optimize thresholds such that each player will have a certain number of deals (e.g., comparable, fewer, larger to others). This can add another layer of complexity that could affect which agent would be more likely to adapt (e.g., agents with more available options can be more flexible in creating coalitions). For example, in the base game, one agent had fewer possible deals and typically that led to that agent being often excluded from the agreement. This also may explain the 5-party agreement rule used in the base game. In our setup, we kept the 5-party agreement while still calculating the 6-party one (in the cooperative and greedy versions) in order to have a meaningful comparison to the adversarial game.
>
> - Thank you for pointing out the relationship to Pareto efficient deals computed based on scores. As you noted, another alternative would be to set the threshold relative to the optimal one or an average of pareto efficient deals’ utilities for the corresponding agent.
> We also now observed that the number of Pareto dominated deals may potentially control the difficulty of games (via changing scores themselves); this may allow rewarding some agents more while keeping the utility at least the same for all other agents, potentially leading to easier agreement. For example, the base game has 239 dominated deals while Game 3 has 569 ones. Although more analysis would be needed to confirm, we empirically found game3 to be easier with also higher 6-way agreement (see also Figure 11 in the appendix that shows the number of agreeing parties per $p_1$'s scores, which was more stable and higher for game3). We will discuss these ideas in the paper.
>
> -----
>
> **- Different number of agents:**
>
> - We show in Table 7 in the appendix experiments with reducing the number of agents.
>
> -----
>
> **- Writing:**
>
> - By “underperform” we meant by comparing the cooperative game to other variants and also in some of the new games. For the p2 “veto” party acting as middle ground, we meant considering its scores wrt p1 and other parties. We generally observed that agents are consistent with their payoffs (e.g., please see Fig.7 in the appendix). We will clarify.

---

### Author Rebuttal · Authors · 2024-08-14

We thank the reviewers for their efforts and time and we genuinely appreciate their suggestions and insights. We are excited that our paper was well-received and we are encouraged by the positive feedback on the paper’s writing and structure, the execution of our benchmark and its potential to dynamically test future models, and the coverage of our experiments and ablation studies.

We reply to some common questions in this global rebuttal and we address individual comments below. We clarify some questions and include additional experiments when needed.

**- Baselines:** (Reviewers XyTr and Dn58)

We agree that computing random and heuristic-based baselines would contextualize the performance of models. We provide baselines either by the statistical properties of games or via simulating randomized interactions.

1- *Random baselines:* We can compute the random chance as a property of scores and thresholds without interactions.

- For each game and difficulty condition, we can statistically compute how likely a purely random deal would lead to success given the thresholds of all parties. For example, for the base game it is 55/720, for the difficulty levels in Table 4, it would be 30/720 and 17/720, respectively. This means that models perform significantly better than random chance.

2- *Interactive baselines:* These are based on repeated interaction or iterations.

- We add a baseline where we prompt LLM agents to give a completely random deal at each round. The results are **10%** and **3%** for 5 and 6 agreements respectively for 120 negotiation sessions. The “wrong deals” are also quite high (~20%). This shows that the reasoning done in the default experiment is quite crucial.

- We add a repeated rule-based baseline that is based on simulating randomized interaction without using LLMs. Here, we start with a random deal and select one agent at a time to improve over the last proposed deal by changing one option at a time until its corresponding minimum threshold is met or no more changes can be made. The agent starts from the highest til the lowest priority issue and for each issue picks the best option. The next agent iterates over the last proposed deal. The last agent to change is p1. We run this for a large number of randomized orders and starting deals and take the unique set of achieved deals and compute 5 and 6 way success ratios.

| Method   | Base game | game 1 | game 2 | game 3 |
| ----------   | --------------- | ---------- | ---------- | ---------- |
| Baseline | 37/28          | 46/22    | 62/20    | 79/28    |
| GPT-4    | 81/33          | 65/10    | 70/40    | 86/81    |
|               |                    |              |              |              |

This also shows that other LLMs fail below the baseline while more capable models outperform it, and it’s also consistent with our analysis that game 3 is the easiest. However, please note that this is a considerably naive baseline as the changes made by an agent are never opposed and applied directly.

-----

**- All-greedy experiment:** (Reviewers XyTr and iQ4e)

- We now add an “all-greedy” experiment for GPT-4. Results obtained (**26%** and **11%** for **5** and **6** agreements respectively) are comparable to “p1 greedy” (**27%** and **9%**) which can be expected as p1 makes the final deals in both scenarios. But we observe interesting qualitative differences when agents point to possible coalitions based on having all parties more clearly and consistently communicating their strongest priorities.

- Thank you for the interesting observation regarding how “greedy” leads to more degradation in reaching agreement compared to “adversarial” variants (Reviewer XyTr). We speculate that this can be due to having the greedy agents’ proposals more aligned with common sense of their roles. In the adversarial game, the proposals are either not coordinated (for the untargeted), or not strongly aligned with that agent’s role, which potentially leads to weaker effect on other agents. This can be an interesting follow-up to study how strongly pretraining common sense is weighed against the counter-intuitive observations and how that can be used to robustify negotiation against adversaries or noisy irrational players.

- We believe this experiment may provide a closer simulation of real-world negotiations and can be used to test the trade-off between cooperation vs. competition (reviewer iQ4e).

-----

**- Scalability to less constrained setups:** (Reviewers XyTr and iQ4e)

- We acknowledge that while the action space is large compared to previous work (due to the number of issues and options) , it still has a constrained format given the options per issues. We will add that to the limitations discussions. Possible future work could rely on exclusively continuous issues rather than discrete (our issues take both continuous, such as budget, and discrete formats, such as locations). For continuous issues, utility can be an arbitrary continuous function.

- In addition, we got preliminary success in increasing the number of issues and options by using GPT-4, indicating that future extensions could be possible.


-----

**- Related work:** (Reviewers XyTr and kvGq)

- We are happy to expand our related work section (which we had to shorten due to space) to give more broader context beyond what we have already covered on LLM negotiation. We will expand the discussion of related work on agents, debate, evaluation, game theory and negotiation in the main paper as much as space allows and further to an appendix if needed.

-----

We will update the paper with these results and discussion points.

---

### Decision · Program_Chairs · 2024-09-26

**Decision:**

Accept (Poster)

**Comment:**

This paper investigates the LLM-based communication and decision-making capabilities multi-agent setups, via a complex multi-agent, multi-issue, and semantically rich negotiation game framework. Scorable negotiation is used to evaluate LLMs. Agents have strong arithmetic, inference, exploration, and planning capabilities, and operate in a dynamic and multi-turn setup. The paper proposes a comprehensive set of metrics to rigorously quantify agents' performance and alignment with their assigned roles. It also defines and implements mechanisms to create new games and increase games' difficulty, for an evolving benchmark. Critical safety aspects such as the interaction dynamics between agents influenced by greedy and adversarial players are evaluated. The paper concludes that GPT-4 and SoTA large models (e.g., Llama-3 70b) still under-perform (when comparing the cooperative game to other variants and also in some of the new games), due to the challenging settings.

The proposed benchmark is highly extensible and flexible. It effectively tests the capabilities of LLM-based agents in inferring others' preferences and combinatorial optimization. The experiments are extensive, and various ablation studies demonstrate the effectiveness of each prompt within the benchmark. Limitations and design decisions were discussed both in the paper and during the rebuttal.

The authors provided detailed and sound responses to the reviewers' comments and suggestions, and it is feasible to include the suggested changes in the final version of the paper.

This paper makes a significant and sound contribution, on a timely topic which is also highly relevant to the Datasets and Benchmarks community.